# THE ILLUSION OF CERTAINTY: UNCERTAINTY QUANTIFICATION FOR LLMS FAILS UNDER AMBIGUITY

## ABSTRACT

Accurate uncertainty quantification (UQ) in Large Language Models (LLMs) is critical for trustworthy deployment. While real-world language is inherently ambiguous, existing UQ methods are typically benchmarked against tasks with no ambiguity. In this work, we demonstrate that while current uncertainty estimators perform well under the restrictive assumption of no ambiguity, they degrade to close-to-random performance on ambiguous data. To this end, we introduce MAQA* and AmbigQA*, the first ambiguous question-answering (QA) datasets equipped with ground-truth answer distributions estimated from factual co-occurrence. We find this performance deterioration to be consistent across different modeling paradigms: using the predictive distribution itself, internal representations throughout the model, and an ensemble of models. We show that this phenomenon can be explained theoretically, revealing that predictive-distribution and ensemble-based estimators are fundamentally limited under ambiguity. Overall, our study reveals a key shortcoming of current UQ methods for LLMs and motivates new approaches that explicitly model uncertainty during training.

## 1 INTRODUCTION

Many linguistic tasks that are solved by Large language models (LLMs) can be framed as *question-answering* (QA): a user poses a query, and the model provides an answer. As LLMs are increasingly deployed in high-stakes domains—such as medical diagnosis, legal advice, or autonomous decision-making it becomes critical not only to obtain correct answers but also to have reliable estimates of how well the model understands the data, also referred to as *epistemic uncertainty*. An important consideration when assessing model reliability in this context is that some questions permit more than one answer. Consider these two examples:

**Single-answer (No ambiguity):** *"Which hormone do I lack if I have type 1 diabetes?"* $\rightarrow$ *Insulin*.

**Multi-answer (Ambiguity):** *"Which medication should I take for type 2 diabetes?"* $\rightarrow$ *Metformin, Sulfonylureas, DPP-4 Inhibitors, ...* (all plausible, but with different probabilities).

Since in the first example there is only one correct answer, any model that predicts a distribution over possible replies should put all mass on this one answer. In the second example, multiple answers are correct, and they may be associated with different probabilities. This is known as *aleatoric uncertainty*: It refers to the randomness that is intrinsic to the distribution of true answers itself. Most uncertainty–quantification (UQ) methods for LLMs, however, are evaluated on data resembling the first question, where aleatoric uncertainty is zero (Devic et al., 2025). In this restrictive setting, a variety of UQ methods show satisfactory performance in estimating *epistemic uncertainty* (Kuhn et al., 2023; Duan et al., 2024; Yadkori et al., 2024). However, many realistic applications involve non-trivial aleatoric uncertainty. This motivates a critical question: *How do current UQ approaches perform under realistic conditions of ambiguity?*

For this, we examine three families of estimators, each exploiting a different source of information: (i) **Predictive Variation**: methods that rely solely on the predictive distribution $p$, typically quantifying epistemic uncertainty via variation measures such as entropy Vashurin et al. (2025). (ii) **Internal Representations**: methods that probe the hidden states of the LLM to infer signals of epistemic uncertainty. (iii) **Ensembles**: Bayesian-inspired methods that approximate a posterior in

Figure 1: *Theoretical Insights on 3-class simplex* **Left:** Under zero aleatoric uncertainty, high entropy guarantees low EU, since all possible $p^*$ are far away (Theorem 1). Assuming a well-trained model, observing a low entropy distribution likely indicates low EU as the model cannot frequently be confidently incorrect (Theorem 2). **Right**: Under non-trivial aleatoric uncertainty, observing high or low entropy does not provide information about the EU, since the ground-truth distribution $p^*$ is not constraint to any particular location in the probability simplex.

the model parameter space by aggregating predictions from multiple models. We demonstrate that all of these methods fail when answers have non-trivial aleatoric uncertainty. In short, we:

- Introduce MAQA$^*$ and AmbigQA$^*$, the first ambiguous QA datasets equipped with explicit ground-truth answer distributions $p^*$, estimated from factual co-occurrence statistics (Section 4). These datasets enable, for the first time, a principled evaluation of uncertainty estimators under real-world ambiguity.

- Empirically confirm that existing methods perform nearly at random in distinguishing and ranking high and low epistemic uncertainty questions when they are inherently ambiguous (Section 5).

- Provide theoretical insights into why variation and ensemble-based methods succeed under zero aleatoric uncertainty (Section 3) but break down once ambiguity is present. (Section 5)

Our findings fundamentally challenge the suitability of existing uncertainty quantification methods for the practical deployment of LLMs. We release our new benchmark with empirical answer distributions to support future research on UQ methods that explicitly account for non-trivial ambiguity already *during model training*.

## 2 BACKGROUND

Uncertainty quantification (UQ) in machine learning (ML) characterizes the uncertainty in a model's predictive distribution for a given input $x$. This uncertainty, often referred to as *total uncertainty*, stems from two distinct sources: *epistemic uncertainty*, reflecting uncertainty in the model itself due to limited training data, model misspecification, or artifacts of optimization, and *aleatoric uncertainty*, which represents intrinsic randomness in the true data-generating process (Hüllermeier & Waegeman, 2021; Gawlikowski et al., 2022). Epistemic uncertainty can be reduced with sufficient data and a well-specified model, whereas aleatoric uncertainty is irreducible by definition. Importantly, when both sources are present, they jointly shape the model's predictive distribution, and naive uncertainty estimates may confuse epistemic uncertainty for genuine data ambiguity. As such, disentangling these sources of uncertainty is a central challenge in reliable ML.

With the general capability of LLMs to address diverse tasks by framing them as question-answering (QA) problems (Sanh et al., 2022), a natural approach to uncertainty quantification in LLMs is assessing the model's certainty in the answers it provides. Since LLMs often produce syntactically diverse yet semantically equivalent answers, it is useful to group answers into semantic equivalence classes (Kuhn et al., 2023). For instance, to the question "What is the capital of France?", the answers "Paris" or "The capital is Paris" represent the same semantic class. We focus on the distribution over these semantically distinct classes, denoted $p$ in the remainder, with implementation details given in Section B. This perspective allows studying uncertainty quantification for LLMs as a classification problem, enabling us to build on established theory.

Following Kotelevskii et al. (2025), we define the *total uncertainty (TU)* as the cross-entropy between the true distribution $p^*$ over semantic classes and the semantic distribution predicted by the model $p$ - in the case of ensembles, $p$ is the model average $\bar{p}$. This allows a natural decomposition: *Aleatoric uncertainty (AU)* is the entropy of the true distribution $p^*$ and *epistemic uncertainty (EU)* the Kullback-Leibler divergence between $p^*$ and predicted distribution $p$[1]:

$$\underbrace{\mathrm{CE}(p^*, p)}_{\text{Total (TU)}} = \underbrace{H(p^*)}_{\text{Aleatoric (AU)}} + \underbrace{\mathrm{KL}(p^*\|p)}_{\text{Epistemic (EU)}} \tag{1}$$

Unlike the widely used information-theoretic decomposition for sampling-based methods Gal et al. (2017); Depeweg et al. (2018), which has faced criticism for conflating distinct sources of uncertainty Wimmer et al. (2023); Smith et al. (2025), this formulation makes use of a reference distribution $p^*$, which is critical for principled evaluation (Smith et al., 2025).

## 2.1 SETUP

**Estimators** We categorize existing estimators into three categories based on the information they use: (i) **Predictive Variation** estimators that are based on the variation of the semantic distribution $p$. We evaluate Semantic Entropy (SE) (Kuhn et al., 2023), Maximum Sentence Probability (MSP), and Shifting Attention to Relevance (SAR) (Duan et al., 2024). While not strictly falling into this category, we additionally test Iterative Prompting (IP) (Yadkori et al., 2024), as it is the only estimator specifically designed for the case of non-trivial AU. (ii) **Internal Representations** estimators that use internal activations throughout the LLM. Here, we extract residual stream activations $h^l$ at layer $l$ for the final input token (pre-generation), and train linear probes and 2-layer MLPs with squared error loss to predict EU. (iii) **Ensemble** estimators that model a Bayesian posterior over the space of models. We use an ensemble of different LLMs to approximate this posterior (Lakshminarayanan et al., 2017) and quantify EU as the Mutual Information (MI) (Depeweg et al., 2018).

**Models** We evaluate the estimators across several models: LLaMA3.1 8B (Grattafiori et al., 2024), Gemma3 12B (Team et al., 2025), Qwen2.5 14B (Qwen et al., 2025)—each in both base and instruct variants. For ensembles, we combine these three architectures, treating them as approximate posterior samples from distinct model classes.

**Metrics** We study how well the estimated EU represents the true EU as quantified in Equation (1). Since both are continuous quantities, our primary evaluation uses the **concordance** statistic $AUC_c$, an estimate of $\mathbb{P}(EU_i > EU_j \mid \text{Estimator}_i > \text{Estimator}_j)$ (Therneau & Atkinson, 2024). It quantifies the probability that the estimator correctly ranks a sample with higher true EU above one with lower true EU in terms of the estimated EU. The resulting score can be interpreted analogously to the traditional AUC-ROC, with 0.5 corresponding to random chance and 1 to perfect ranking. For additional experiments, we also report **AUC-ROC**, where for a given threshold $\delta$ we measure the separation between uncertain ($EU \geq \delta$) and certain ($EU < \delta$) samples.

## 3 WHEN CURRENT UQ WORKS: ZERO ALEATORIC UNCERTAINTY

We first revisit the zero-AU setting. Nearly all prior work evaluates UQ methods (Devic et al., 2025) under this assumption, and consequently, estimators perform well. We confirm this observation on the unambiguous factual question answering dataset TriviaQA[2](Table 1).

We hence ask if theoretical insights can explain their success? While the effectiveness of internal representation methods remains largely empirical, estimators relying on predictive variation and ensembles admit a more principled theoretical interpretation that reveals useful structure. Our theoretical explanation for the success of these methods relies on the insight that if AU is zero, the EU reduces to the negative log-probability the model assigns to the correct semantic class: $EU = -\log p(y = y^*)$(see Proposition 3). Therefore, the EU can be directly understood as the model's confidence in the correct answer. Visually, this means that the true distribution $p^*$ must be located in one of the vertices of the probability simplex Figure 1. Based on this insight, we derive two complementary results for estimators based on predictive variation and ensembles.

---

[1]We assume that the model class is sufficiently expressive to represent $p^*$; hence, all mismatch between $p$ and $p^*$ can in principle be reduced

[2]Note that we use the first 2000 samples as this is sufficient to demonstrate our case

Table 1: Concordance scores $AUC_c$ for all estimators on TriviaQA (AU=0)

| Model | Predictive Var. | | | | Internal Rep. | | Ensemble |
|---|---|---|---|---|---|---|---|
| | SE | MSP | SAR | IP | Linear | MLP | MI |
| Llama 3.1-8B | 0.80 | 0.74 | 0.79 | 0.80 | 0.66 | 0.71 | 0.85 |
| Gemma 3-12B | 0.91 | 0.79 | 0.86 | 0.90 | 0.66 | 0.73 | 0.85 |
| Qwen 2.5-14B | 0.87 | 0.74 | 0.82 | 0.86 | 0.65 | 0.69 | 0.85 |
| Llama 3.1-8B-Instruct | 0.84 | 0.79 | 0.83 | 0.81 | 0.68 | 0.73 | 0.87 |
| Gemma 3-12B-Instruct | 0.76 | 0.76 | 0.77 | 0.74 | 0.72 | 0.79 | 0.87 |
| Qwen 2.5-14B-Instruct | 0.73 | 0.69 | 0.73 | 0.69 | 0.75 | 0.80 | 0.87 |

## 3.1 WHY PREDICTIVE VARIATION IS INFORMATIVE UNDER ZERO AU

Predictive variation-based methods rely on variation in the predictive distribution $p$. Focusing on predictive entropy $H(p)$ as our central example, we can establish both a lower bound in the high-entropy case and a probabilistic upper bound in the low-entropy case. Importantly, the corresponding insights translate to other variability-based uncertainty measures as well.

**Theorem 1** (High Entropy $\Rightarrow$ High EU). *Let there be $K \geq 2$ classes and $\delta \in [0, \log K]$ be a threshold on the entropy indicating uncertainty. Furthermore, let $\alpha_\delta$ be the maximal possible probability on some class s.t. $H(p) \geq \delta$. Then the epistemic uncertainty with $H(p) \geq \delta$ is at least:*

$$EU = \mathrm{KL}(p^* \| p) \geq -\log \alpha_\delta.$$

Intuitively, a high entropy $H(p) \geq \delta$ implies that the predictive distribution must become increasingly less concentrated on the probability simplex. This implies that the maximum probability assigned to any class can be at most $\alpha_\delta$ - naturally, also for the correct class $y^*$. Since epistemic uncertainty is quantified as $-\log p(y = y^*)$, such a flat predictive distribution hence leads to large epistemic uncertainty (Figure 1). Thus, Theorem 1 explicitly shows that *high predictive entropy necessarily implies high epistemic uncertainty*.

**Theorem 2** (Low Entropy $\Rightarrow$ Low EU with High Probability). *Let there be $K \geq 2$ classes and $\delta \in [0, \log 2]$ be a threshold on the entropy indicating uncertainty. Furthermore let $\bar{\mathcal{L}} = \mathbb{E}_{(x,y)}[-\log p_y]$ be the model's average loss and $\gamma_\delta$ be the minimal maximal confidence in a prediction $p$ s.t $H(p) \leq \delta$. Then the probability that the epistemic uncertainty with $H(p) \leq \delta$ will be less than $-\log(\gamma_\delta)$ satisfies:*

$$\mathbb{P}(EU \leq -\log(\gamma_\delta) \mid H(p) \leq \delta) \geq 1 - \frac{\bar{\mathcal{L}}}{-\log(1 - \gamma_\delta) * \mathbb{P}(H(p) \leq \delta)}$$

Theorem 2 complements Theorem 1 by showing that low entropy likely implies low epistemic uncertainty. When the predictive entropy is small, i.e. $H(p) \leq \delta$, most of the probability mass must lie on a single class with weight at least $\gamma_\delta$. This induces a dichotomy: if the class is correct, epistemic uncertainty is small ($\leq -\log \gamma_\delta$); if incorrect, it is large ($\geq -\log(1 - \gamma_\delta)$) (Figure 1). A deterministic upper bound is therefore impossible, but we can obtain a probabilistic guarantee that depends on the model's performance. Noting that the training loss $-\log p_y$ coincides with epistemic uncertainty under zero AU (Proposition 3), the average loss $\mathcal{L}$ is the expected EU. For a well-trained model with small $\mathcal{L}$, frequent high-EU errors are hence unlikely to occur. In other words, highly confident but incorrect predictions must occur only rarely. The bound depends on the probability of the model making confident predictions, which means it may become loose if such cases are very rare, since their contribution to the average loss is then negligible. In practice, however, models are trained toward confident predictions, making such cases unlikely. Put differently, Theorem 2 shows that for models that are likely to make confident predictions and perform well on average, *observing a high predictive entropy corresponds likely to a low epistemic uncertainty*.

## 3.2 WHY ENSEMBLE-BASED UQ WORKS UNDER ZERO AU

Showing that zero AU enables predictive entropy to be a reliable estimate of the true epistemic uncertainty has direct implications for ensemble-based UQ. For ensembles, EU is quantified as the

Table 2: Examples of question-answer-distribution pairs

| Dataset | Question | Answer(s) | # Counts in Data | $p^*$ | Entropy $H(p^*)$ |
|---------|----------|-----------|------------------|-------|------------------|
| TriviaQA | Where in England was Dame Judi Dench born? | $\{Yorkshire\}$ | n/a | $[1.00]$ | 0.0 |
| MAQA* | What is one essential component of the fire triangle? | $\{Heat, Fuel, Oxygen\}$ | $\{31, 32, 25\}$ | $[0.35, 0.36, 0.29]$ | 1.1 |
| AmbigQA* | What is the name of one princess in Frozen? | $\{Elsa, Anna\}$ | $\{188, 91\}$ | $[0.67, 0.33]$ | 0.63 |

mutual information (MI) between the model parameters and the predicted target variable.

$$\underbrace{\mathrm{MI}(\bar{p}; \theta)}_{\text{Estimated EU}} = H(\bar{p}) - \mathbb{E}_\theta\left[H(p_\theta)\right] \leq H(\bar{p}), \tag{2}$$

where $\bar{p} = \mathbb{E}_\theta\left[p_\theta\right]$ is the Bayesian model average that serves as the ensemble's prediction. Equation (2) shows that the MI, which estimates EU, is bounded by the entropy of the ensemble's predictive distribution. Therefore, a large MI implies a large entropy in $\bar{p}$ which, in turn, implies high true epistemic uncertainty as per Theorem 1. Thus, mutual information is not merely an empirical heuristic but admits a theoretical justification in this case: *in the zero-AU setting, large mutual information necessarily signals high true epistemic uncertainty.*

Similarly to prediction-based estimators, low mutual information does not guarantee a low epistemic uncertainty. However, if individual predictors $p_\theta$ achieve low expected error, then most members assign high probability mass to the correct label, resulting in near-zero entropy predictions. As such $\mathbb{E}_\theta[H(p_\theta)] \approx 0$ and by Equation (1), this gives $\mathrm{MI}(\bar{p}; \theta) \approx H(\bar{p})$. Thus, MI closely tracks the entropy of the model average. Moreover, by Jensen's inequality,

$$-\log \bar{p}(y^\star) = -\log \mathbb{E}_\theta[p_\theta(y^\star)] \leq \mathbb{E}_\theta[-\log p_\theta(y^\star)],$$

implying that if the individual models are accurate on average, the average model prediction is accurate as well. Therefore, we can apply Theorem 2 again to conclude that low MI likely corresponds to low entropy in $\bar{p}$, which in turn corresponds to low epistemic uncertainty.

**Takeaway** The zero-AU case paints a consistent picture: all estimators provide faithful estimates. Since the true EU reduces to the negative log-likelihood, this behavior can be theoretically explained for both prediction-variation and ensemble-based estimators. Critically, these arguments rely on the absence of aleatoric uncertainty. Yet real language tasks rarely satisfy this condition, as ambiguity is inherent to language. This necessitates the design of novel benchmarks that evaluate epistemic uncertainty estimation under non-zero aleatoric uncertainty.

## 4 A Novel QA benchmark for non-zero Aleatoric Uncertainty

While ambiguous QA datasets such as MAQA Yang et al. (2025) and AmbigQA Min et al. (2020) exist, none provide ground-truth answer distributions $p^*$, which makes it impossible to quantify the true epistemic uncertainty $EU = \mathrm{KL}(p^*\|p)$. We close this gap and introduce MAQA* & AmbigQA*, which for the first time enables a systematic quantitative evaluation of UQ methods under realistic ambiguity.

### 4.1 Approximating $p^*$ via Corpus Statistics

To approximate $p^*$, we assume a frequentist view: the probability of an outcome should equal its relative frequency in the (pre-)training data distribution. Concretely, for a question $x$ and candidate answer $y_i$, we approximate $p^*(y_i \mid x)$ by the rate at which the underlying *fact* occurs in the pretraining corpus. For example, if the statement "Metformin is a medication for type 2 diabetes" appears more often than "Sulfonylureas is a medication for type 2 diabetes," then $p^*(\text{Metformin} \mid x) \geq p^*(\text{Sulfonylureas} \mid x)$. This choice is well supported by previous work: Empirically, co-occurrence statistics correlate strongly with model performance: models score higher on samples with frequent co-occurrence Kandpal et al. (2023); Mallen et al. (2023), and recently, Wang et al. (2025)demonstrate that, particularly in factual QA, LLM output probabilities correlate with co-occurrence statistics. Theoretically, estimating $p^*$ from corpus statistics is more principled than relying on external annotations: crowd-sourced labels reflect the annotator distribution rather than the pretraining distribution $p_{\text{train}}$. As the number of training samples $n \to \infty$, epistemic uncertainty vanishes while aleatoric ambiguity persists Smith et al. (2025), so the output distribution of a well-trained model should converge to $p_{\text{train}}$.

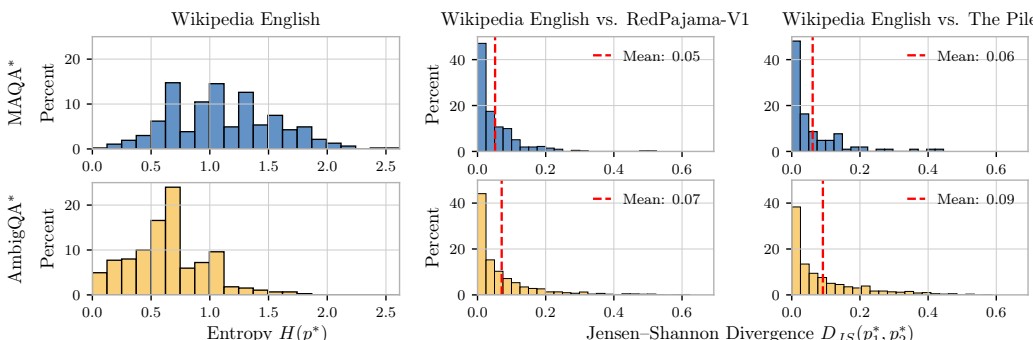

Figure 2: **Left**: Distribution of ground-truth entropy $H(p^*)$ across questions in MAQA$^*$ and AmbigQA$^*$, **Right**: Distribution of JS divergences between different proxys for estimating $p^*$. The low divergence validates the quality of these distributions.

## 4.2 OBTAINING THE TRUE DISTRIBUTION $p^*$

Since the pre-training datasets for LLMs are not publicly available, we instead employ the English Wikipedia Wikimedia Enterprise (2024) as a proxy for the pre-training corpus due to its widespread use in LLM pre-training and comprehensive coverage of factual knowledge. To perform the co-occurrence search, we use keywords extracted from the question alongside candidate answers. The keywords represent the most important words in the question, e.g., the question's subject, and importantly, both keywords and answers are stemmed to their base forms to ensure robustness against surface-form variation. Elsahar et al. (2018) demonstrate that subject–object co-occurrence is a reliable indicator for the presence of a subject–relation–object triplet, making it suitable for fact counting. We further improve the precision of these counts by using an entailment model to verify the factual occurrence of each candidate co-occurrence. The resulting datasets contain 468 and 2553 Q&A examples, respectively. Their semantic answer-entropy distributions (Figure 2) span a diverse range of true distributions $p^*$, with examples shown in Table 2.

We validate the counts obtained through this method by comparing them to distributions estimated from two alternative co-occurrence counting strategies: (i) Similarly as above, using keywords and answers but using as corpus the RedPajama-V1 dataset (Weber et al., 2024) via infini-gram (Liu et al., 2024), and (ii) through entity linking on the Pile dataset (Gao et al., 2020) using DPBedia Spotlight (Kandpal et al., 2023; Daiber et al., 2013). We find that the distributions obtained from all strategies align closely, with Jensen–Shannon divergences between the estimated ground truths $p^*$ being small in most cases (Figure 2). This consistency validates the quality of our constructed ground-truth distributions $p^*$ (Section C).

## 5 NON-TRIVIAL ALEATORIC UNCERTAINTY

Using our novel datasets with non-trivial aleatoric uncertainty and ground truth probabilities, we investigate how UQ approaches for LLMs perform under ambiguity. Overall, we find that performance clearly collapses, as seen in Table 3. Methods that performed well in the zero-AU setting perform only marginally better than random chance under ambiguity. This pattern is consistent across prediction-based, representation-based, and ensemble-based estimators and across all model families. We further validate our findings for the family of prediction-based estimators in Section A.1, showing that the results also hold under alternative strategies for estimating $p^*$ and across different model sizes.

These observations raise a central question: *why do seemingly robust estimators fail once AU is non-trivial?* We investigate the shortcomings of these UQ methods.

### 5.1 LIMITATIONS OF PREDICTION-BASED ESTIMATORS

In Section 3.1, we show that under zero aleatoric uncertainty, high entropy indicates epistemic uncertainty, whereas low entropy predominantly likely reflects epistemic confidence. These insights leverage the fact that under zero aleatoric uncertainty, the ground-truth is constrained to be an indi-

Table 3: Concordance scores $AUC_c$ for all estimators on MAQA and AmbigQA.

| Model | MAQA | | | | | | | AmbigQA | | | | | | |
| | Predictive Var. | | | Internal Rep. | | Ensemble | Predictive Var. | | | | Internal Rep. | | Ensemble |
| | SE | MSP | SAR | IP | Linear | MLP | MI | SE | MSP | SAR | IP | Linear | MLP | MI |
|---|---|---|---|---|---|---|---|---|---|---|---|---|---|---|
| Llama 3.1-8B | 0.52 | 0.49 | 0.51 | 0.53 | 0.57 | 0.57 | 0.51 | 0.61 | 0.58 | 0.60 | 0.60 | 0.57 | 0.61 | 0.58 |
| Gemma 3-12B | 0.55 | 0.53 | 0.58 | 0.60 | 0.62 | 0.59 | 0.51 | 0.66 | 0.64 | 0.66 | 0.66 | 0.56 | 0.62 | 0.58 |
| Qwen 2.5-14B | 0.59 | 0.56 | 0.62 | 0.59 | 0.65 | 0.64 | 0.51 | 0.67 | 0.63 | 0.67 | 0.66 | 0.59 | 0.62 | 0.58 |
| Llama 3.1-8B-Instruct | 0.54 | 0.52 | 0.53 | 0.54 | 0.59 | 0.56 | 0.50 | 0.60 | 0.59 | 0.60 | 0.60 | 0.57 | 0.61 | 0.60 |
| Gemma 3-12B-Instruct | 0.53 | 0.54 | 0.55 | 0.54 | 0.57 | 0.56 | 0.50 | 0.57 | 0.57 | 0.58 | 0.57 | 0.57 | 0.62 | 0.60 |
| Qwen 2.5-14B-Instruct | 0.55 | 0.54 | 0.56 | 0.55 | 0.62 | 0.60 | 0.50 | 0.57 | 0.56 | 0.57 | 0.56 | 0.58 | 0.62 | 0.60 |

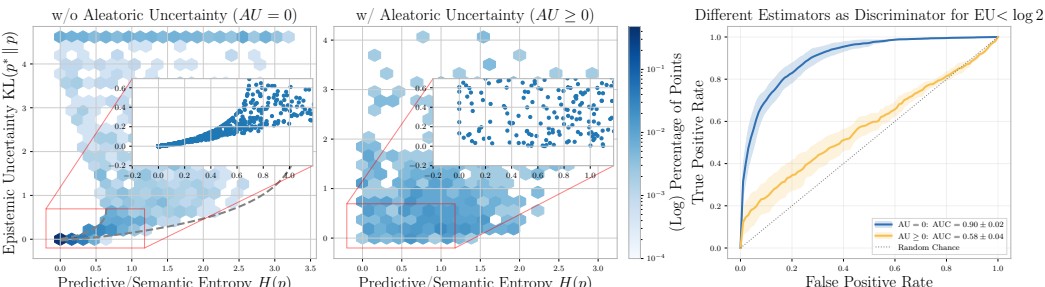

Figure 3: *Relationship between prediction-based estimators and true epistemic uncertainty (EU) for Gemma 3-12B on MAQA*. **Left**: Relationship between $H(p)$ and true EU. If aleatoric uncertainty (AU) is zero predictive entropy and prediction-based EU correlate. This correlation vanishes under non-trivial AU. Lines indicate theoretical bounds on EU [3]. **Right** The average ROC curve of prediction-based estimators for identifying predictions with high true EU (EU $< \log(2)$) approaches random performance. Shaded regions represent one standard deviation over different estimators.

cator distribution and must be located in one of the vertices of the probability simplex. Allowing for aleatoric uncertainty lifts this restriction on $p^*$. Consequently, a high-entropy prediction no longer necessarily indicates high EU as the entropy may also arise from an inherently uncertain ground-truth (Figure 1 right) that is, at the same time, well reflected by the model (low EU). More generally, we can show that no function of the predictive distribution $p$ alone can distinguish epistemic uncertainty from intrinsic ambiguity:

**Proposition 1** (Non-Identifiability of Epistemic Uncertainty). *Let $K \geq 2$ and $\Delta^{K-1}$ be the probability simplex over $K$ classes. For any function $f : \Delta^{K-1} \to \mathbb{R}$ and any $p \in \Delta^{K-1}$, there exist $p_1^*, p_2^* \in \Delta^{K-1}$ such that*

$$\text{KL}(p_1^* \,\|\, p) = 0 \quad and \quad \text{KL}(p_2^* \,\|\, p) = -\log \min_i p_i \geq \log K,$$

*Thus, the model's prediction $p$, and consequently any function $f(p)$, can both indicate zero epistemic uncertainty or high epistemic uncertainty ($\geq \log(K)$).*

As such, any estimator that is a function of $p$—e.g. semantic entropy—cannot faithfully estimate EU without restrictions on AU. Empirically, the contrast between the two cases can be seen in Figure 3: If AU is zero, Theorem 1 lower-bounds the entropy and ensures that predictions with sufficient entropy cannot correspond to low true epistemic certainty. The primary sources of errors in the zero AU case are confident yet incorrect predictions (left top). However, given a sufficiently well-trained model, these occur with low probability (Theorem 2), which is reflected in the sparsity of that region (low bin counts). Conversely, for non-trivial AU, predictive entropy has no connection to EU. Pathological cases include predictions with high predictive entropy despite low EU, which can be seen in the $AU \geq 0$ case of Figure 3 (middle plot) in the right/middle bottom section. Another case are predictions exhibiting higher EU but low predictive entropy, which are located on the left-middle top section.

---

[3]The Lower bound is based on $K = 30$ and could be significantly sharper for fewer classes

## 5.2 LIMITATIONS OF ENSEMBLES-BASED ESTIMATORS

Because of the strong dependence of mutual information as an estimator of EU and the entropy of the ensemble prediction $\bar{p}$ (see Equation (2)), Proposition 1 has immediate consequences for ensemble-based epistemic UQ as well.

**Proposition 2** (High MI $\not\Rightarrow$ High EU). *Let $K \geq 2$ and $\Delta^{K-1}$ be the probability simplex over $K$ classes. Let $\delta \in [0, \log K]$ be an arbitrary threshold on MI indicating uncertainty. Let $p_\theta$ be such that $\mathrm{MI}(\bar{p}; \theta) > \delta$ with $\bar{p} = \mathbb{E}_\theta [p_\theta]$. Then $p^* = \bar{p} \in \Delta^{K-1}$ results in true epistemic uncertainty $\mathrm{KL}(p^* \parallel \bar{p}) = 0$.*

Intuitively, Proposition 2 stands in direct opposition to Section 3.2: In the zero AU case, high MI implied a high distance from the true $p^*$ that must be located in a corner of the probability simplex $\Delta^{K-1}$. Lifting this restriction, for *any* $\bar{p}$ the true distribution $p^* = \bar{p}$ is associated with zero true EU no matter its associated MI (which is upper bounded by the entropy). Therefore, MI cannot also reliably indicate high EU.

## 5.3 LIMITATIONS OF INTERNAL REPRESENTATIONS

We have empirically and theoretically shown that relying on the predictive distribution of one or more models cannot faithfully estimate the EU under ambiguity. Representing a model's knowledge through a (set of) predictive distribution(s) may collapse signals encoded in the model's internal representations that are relevant to UQ. Therefore, we also investigate linear and MLP-based probes on the model's residual stream as a predictor of EU which is an effective strategy under the absence of AU (Table 1).

Figure 4 shows that the probe performance across different layers degrades under non-zero AU. This indicates that the model's hidden representations contain no additional signal to quantify EU beyond what is already encoded in the predictive distribution.

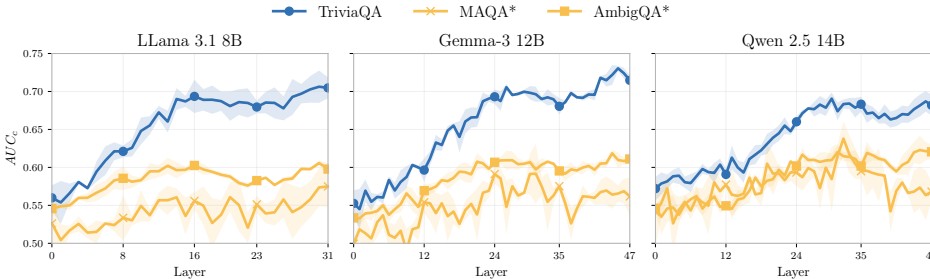

Figure 4: **MLP regression performance across layers**. Under zero AU, probes achieve satisfactory ranking capability in deeper layers. Under non-trivial AU, performance collapses significantly, showing that hidden states do not reliably encode EU when ambiguity is present.

**Takeaway**  All estimators for EU deteriorate greatly under ambiguity, with prediction and ensemble-based methods provably being conceptually flawed. No estimate significantly outperforms a random baseline. This highlights ambiguity as a key gap in the current literature that current methods cannot effectively overcome.

## 6 RELATED WORK

**UQ for LLMs**  A wide range of methods for uncertainty quantification in LLMs have been proposed (Vashurin et al., 2025; Liu et al., 2025). Many methods rely on the predictive distribution $p$. The most prominent approaches here quantify the variation in $p$, with Semantic Entropy (Kuhn et al., 2023) being the most widely adopted, alongside variants such as Duan et al. (2024); Nikitin et al. (2024). In contrast, other methods access model internals. Prior work has shown that hidden states can encode factual correctness (Li et al., 2023; Chen et al., 2024; Orgad et al., 2025). However, to our knowledge, no work has directly investigated if representations provide a reliable signal to estimate epistemic uncertainty itself. Lastly, ensemble methods, which approximate a sample from

the posterior over model weights by training multiple models are often regarded as the gold standard for UQ in classical ML (Lakshminarayanan et al., 2017). While conceptually appealing, their application to LLMs is constrained by prohibitive computational cost and often limited fine-tuning (Balabanov & Linander, 2025).

**Ambiguity in QA Tasks** Previous work are benchmarked on QA datasets like TriviaQA (Joshi et al., 2017) which only contain a single correct answer per question (Devic et al., 2025). Few works consider the presence of aleatoric uncertainty. Hou et al. (2024) introduce aleatoric uncertainty through ambiguity in the question's phrasing. Crucially, this does not cover the case where the ambiguity is inherent to the answer. Yadkori et al. (2024) proposes a that an epistemically confident model should be less likely to be misled by the inclusion of a wrong answer in the input context. Our results show that this is ineffective under ambiguity as well.

The absence of evaluations under ambiguity is a consequence of the lack of suitable benchmarks. Only few datasets explicitly consider ambiguous questions, namely AmbigQA (Min et al., 2020) and MAQA Yang et al. (2025). To our knowledge, MAQA is the only dataset with questions for which we ambiguity is inherent to the task and can not be resolved with a more precise phrasing. It, however, does not quantify the true distribution $p^*$ and, therefore, can not be used for a quantitative study on UQ under ambiguity.

## 7 DISCUSSION

**Limitations** Our new benchmark quantifies $p^*$ as factual occurrences in Wikipedia. Although evidence suggests that such occurrences correlate well with model performance (Kandpal et al., 2023; Mallen et al., 2023; Wang et al., 2025), there is, to our knowledge, no work that empirically shows LLMs approach this distribution in the infinite data limit. We aim to mitigate potential inaccuracies by experimentally verifying for prediction-based estimators the robustness of our findings under Dirichlet-distributed perturbations around the estimated ground truth $p^*$ (Section A.1.1). Furthermore, the deterioration of methods based on internal model representations is purely empirical, and we leave a theoretical analysis to this broader family of approaches to future work. Nevertheless, the empirical evidence for this paradigm is consistent across all models, and additional experiments using classification probes of these representations support the conclusion we arrive at (see Section A.2). Lastly, our evaluation phrases UQ for LLMs as a classification problem and therefore requires models to provide a single answer for each question. While this is consistent with prior work (Kuhn et al., 2023; Aichberger et al., 2024a;b), settings in which multiple answers are generated simultaneously require a fundamentally different theoretical framework for modeling uncertainty.

**Current Estimators are not reliable** With our novel benchmark that spans a wide range of aleatoric uncertainty distributions, we demonstrate that, in general, the performance of epistemic uncertainty estimators collapses under ambiguity. For prediction- and ensemble-based methods, this shortcoming is further supported by theoretical insights. This highlights a systematic flaw in most current UQ methods. Consequently, applying these estimators in general language tasks is problematic and necessarily unreliable.

**Toward Reliable Estimators** Our study shows that none of the common UQ paradigms (prediction-based, internal representations, ensembles) are reliable estimators in the presence of aleatoric uncertainty. Notably, all these paradigms are applied post-hoc to models that are not explicitly trained to maintain or encode uncertainty in their predictions. Hence, a natural next step is to account for uncertainty during training itself. For example, in classical UQ, evidential deep learning (Sensoy et al., 2018) learns a second-order distribution over predictive distributions to represent epistemic uncertainty. More recent work trains models on paired responses to expose epistemic uncertainty (Johnson et al., 2024), or, in the domain of LLMs, proposes reward functions that jointly optimize correctness and model calibration (Damani et al., 2025). We believe that our empirical and theoretical findings guide the development of new estimators toward rethinking the paradigms that are currently employed. Our novel benchmark supports this effort by enabling a more comprehensive evaluation of UQ in LLMs.

ETHICS STATEMENT

In this work, we examine how well large language models can assess their own confidence in a prediction. While any research may be misused, our primary goal is to improve the reliability of these models to support their safe deployment in critical domains. We believe the benefits will outweigh the potential risks.

REPRODUCIBILITY STATEMENT

Our contributions are twofold. First, we theoretically demonstrate that these techniques fail under non-zero aleatoric uncertainty; full proofs are provided in Appendix D. Second, we empirically validate these findings, constructing new datasets for evaluation. The datasets and the code to reproduce all experiments will be released upon acceptance.

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

# A   ADDITIONAL EXPERIMENTS

## A.1   PREDICTIVE VARIATION ESTIMATORS

### A.1.1   ACCOUNTING FOR UNCERTAINTY IN ESTIMATING $p^*$

In practice, our estimate of the ground-truth distribution $p^*$ is itself uncertain due to limited or noisy co-occurrence counts. To explicitly capture this uncertainty, we use a Dirichlet prior $p^* \sim \text{Dir}(\alpha)$, with parameters $\alpha = (\alpha_1, \ldots, \alpha_C)$. We start with a uniform prior $\alpha_i = 1$ for all classes $i$. After observing co-occurrence counts $n_i$, the posterior parameters become $\alpha_i = 1 + n_i$. To prevent low-count posteriors from remaining too uniform—which would erroneously decouple the model prediction $p$ from $p^*$—we introduce a scaling factor $\gamma \geq 1$, defining

$$\alpha_i = 1 + \gamma\, n_i.$$

Then, under the Dirichlet posterior, the *aleatoric uncertainty* is given by:

$$\mathbb{E}_{p^* \sim \text{Dir}(\alpha)}\big[H(p^*)\big] = \mathbb{E}_{p^* \sim \text{Dir}(\alpha)}\big[-\sum_{i=1}^{C} p_i^* log(p_i^*)\big]$$

$$= -\sum_{i=1}^{C} \mathbb{E}_{p^* \sim \text{Dir}(\alpha)}[p_i^* log(p_i^*)]$$

$$= -\sum_{i=1}^{C} [\frac{\alpha_i}{\alpha_0}(\psi(\alpha_i + 1) - \psi(\alpha_0 + 1)]$$

where $\psi$ is the digamma function, and we leverage the fact that each $p_i^* \sim Beta(\alpha_i, \alpha_0 - \alpha_i)$. Likewise, the *epistemic uncertainty* is defined as

$$\mathbb{E}_{p^* \sim \text{Dir}(\alpha)}\big[KL(p^* \| p)\big] = \mathbb{E}_{p^* \sim \text{Dir}(\alpha)}\big[CE(p^* \| p)\big] - \mathbb{E}_{p^* \sim \text{Dir}(\alpha)}\big[H(p^*)\big]$$

$$= -\sum_{i=1}^{C} \mathbb{E}_{p^* \sim \text{Dir}(\alpha)}[p_i^*] log(p_i) - \mathbb{E}_{p^* \sim \text{Dir}(\alpha)}\big[H(p^*)\big]$$

$$= -\sum_{i=1}^{C} \frac{\alpha_i}{\alpha_0} log(p_i) + \sum_{i=1}^{C} [\frac{\alpha_i}{\alpha_0}(\psi(\alpha_i + 1) - \psi(\alpha_0 + 1)$$

$$= \sum_{i=1}^{C} \frac{\alpha_i}{\alpha_0} \big[(\psi(\alpha_i + 1) - \psi(\alpha_0 + 1) - log(p_i)\big]$$

We perform ablation studies over different values of $\gamma$ (see Table 4). Increasing $\gamma$ corresponds to making a stronger assumption that the retrieved $p^*$ is exact, which causes the concordance score to approach the values reported in our main results. For smaller $\gamma$, $p^*$ becomes more independent of $p$, especially given the relatively low counts noted earlier. Interestingly, estimator performance degrades further when we relax the assumption that $p^*$ is exact, corroborating our main findings.

Table 4: Concordance scores $AUC_c$ for Gemma 3-12B for different likelihood multipliers ($\gamma$) across uncertainty estimators.

| Likelihood Multiplier ($\gamma$) | MAQA* | | | | AmbigQA* | | | |
|---|---|---|---|---|---|---|---|---|
| | SE | MSP | SAR | IP | SE | MSP | SAR | IP |
| $\gamma = 1$ | 0.50 | 0.50 | 0.53 | 0.54 | 0.56 | 0.57 | 0.57 | 0.56 |
| $\gamma = 2$ | 0.51 | 0.51 | 0.54 | 0.56 | 0.58 | 0.58 | 0.59 | 0.58 |
| $\gamma = 5$ | 0.53 | 0.52 | 0.55 | 0.57 | 0.61 | 0.60 | 0.61 | 0.61 |
| $\gamma = 10$ | 0.54 | 0.52 | 0.56 | 0.58 | 0.62 | 0.61 | 0.63 | 0.62 |
| $\gamma = 100$ | 0.55 | 0.53 | 0.57 | 0.59 | 0.65 | 0.63 | 0.65 | 0.65 |
| Main KL | 0.55 | 0.53 | 0.58 | 0.60 | 0.66 | 0.64 | 0.66 | 0.66 |

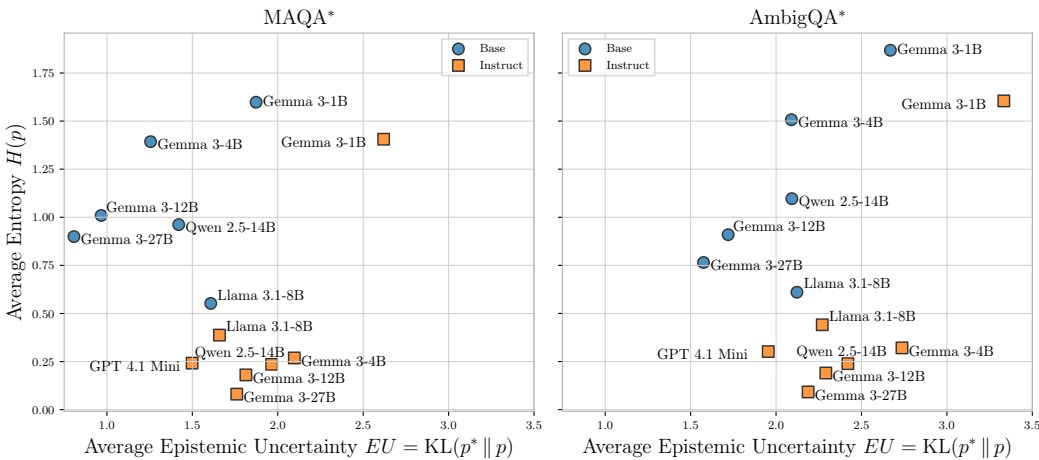

Figure 5: Entropy collapse of Instruct models on MAQA$^*$ and AmbigQA$^*$

### A.1.2 DIFFERENT $p^*$ ESTIMATION METHODS

We assess the robustness of our results by evaluating different strategies for estimating the ground-truth $p^*$, as outlined in Section C. Across all estimators, the three methods yield highly similar results (Table 5), consistent with our observation that their estimated ground truths are strongly aligned. Note that, since we discard samples where at least one class has zero counts, different estimation strategies result in slightly different final datasets.

Table 5: Concordance scores $AUC_c$ for Gemma 3-12B for different estimation methods for ground truth $p^*$

| $p^*$ **Estimation Method** | **MAQA**$^*$ | | | | **AmbigQA**$^*$ | | | |
|---|---|---|---|---|---|---|---|---|
| | **SE** | **MSP** | **SAR** | **IP** | **SE** | **MSP** | **SAR** | **IP** |
| Wikipedia English | 0.55 | 0.53 | 0.58 | 0.60 | 0.66 | 0.64 | 0.66 | 0.66 |
| RedPajama-V1 | 0.55 | 0.53 | 0.58 | 0.59 | 0.65 | 0.63 | 0.65 | 0.65 |
| The Pile | 0.53 | 0.53 | 0.58 | 0.58 | 0.60 | 0.58 | 0.60 | 0.61 |

### A.1.3 INSTRUCT MODELS ENTROPY COLLAPSE

For instruct models, an additional insight is that the entropy for instruct models collapses to zero for most samples, even in cases with non-trivial aleatoric uncertainty. This behavior is undesirable, as it indicates that the models fail to represent any meaningful predictive distribution. Compared to base models, this collapse results in substantially worse model performance (average EU) (Figure 5). Moreover, the entropy collapse also degrades estimator performance on TriviaQA, since a model that always outputs a single answer provides no variability and thus no basis to distinguish certain from uncertain cases.

### A.1.4 EFFECT OF MODEL SIZE

We evaluate different versions of Gemma 3—1B, 4B, 12B, and 27B—and observe that smaller models yield better performance for UQ estimation methods using variation of the predictive distribution (Table 6) on MAQA$^*$. This effect appears to stem from the fact that smaller models often do not know the correct answers, and thus produce arbitrary outputs that form a high-entropy distribution. Such cases naturally coincide with high epistemic uncertainty, as the model lacks knowledge of the answers. Conversely, when a smaller model does know the answers, the resulting distribution has lower entropy and correspondingly lower epistemic uncertainty. As shown in Figure 5, the average entropy decreases substantially with model size. Crucially, this reduction is accompanied by improved performance, indicating that larger models more accurately capture the underlying ground-

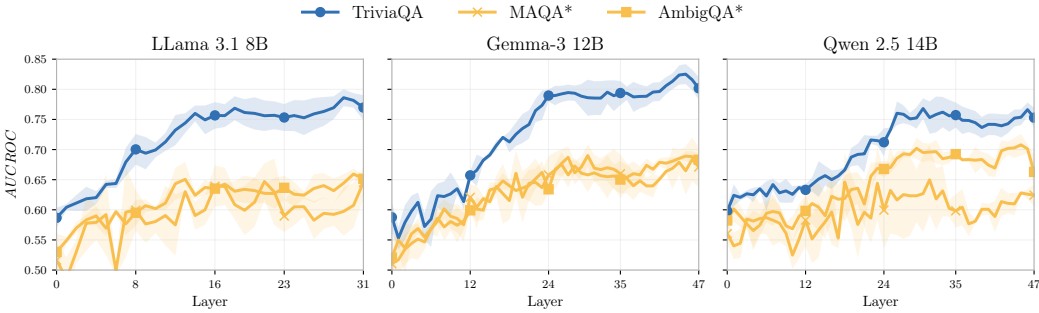

Figure 6: **MLP classification performance across layers**. Under zero AU, probes achieve satisfactory separation capability in deeper layers. Under non-trivial AU, performance collapses significantly, showing that hidden states do not reliably encode EU when ambiguity is present.

truth distributions. The reduced estimator performance of smaller models on TriviaQA is consistent with prior observations in the literature (Kuhn et al., 2023).

Table 6: Concordance scores $AUC_c$ for all estimators of different model sizes on TriviaQA (AU=0) and on AmbigQA$^*$ & MAQA$^*$ (AU≥0). An $AUC_c = 0.50$ corresponds to random chance.

| Model | AU = 0 | | | | AU ≥ 0 | | | | | | | |
| | **TriviaQA** | | | | **MAQA$^*$** | | | | **AmbigQA$^*$** | | | |
| | SE | MSP | SAR | IP | SE | MSP | SAR | IP | SE | MSP | SAR | IP |
|---|---|---|---|---|---|---|---|---|---|---|---|---|
| Gemma 3-1B | 0.78 | 0.72 | 0.77 | 0.78 | 0.69 | 0.63 | 0.71 | 0.69 | 0.67 | 0.64 | 0.64 | 0.64 |
| Gemma 3-4B | 0.85 | 0.76 | 0.82 | 0.85 | 0.65 | 0.57 | 0.65 | 0.67 | 0.69 | 0.65 | 0.69 | 0.67 |
| Gemma 3-12B | 0.91 | 0.79 | 0.86 | 0.90 | 0.55 | 0.53 | 0.58 | 0.60 | 0.66 | 0.64 | 0.66 | 0.66 |
| Gemma 3-27B | 0.93 | 0.80 | 0.87 | 0.91 | 0.52 | 0.52 | 0.56 | 0.57 | 0.65 | 0.62 | 0.65 | 0.64 |

### A.1.5 AUCROC FOR DIFFERENT UNCERTAINTY THRESHOLDS $\delta$

For completeness, we also report AUCROC scores for thresholds $\delta$ other than $\log(2)$ across all datasets (Table 7).The higher values observed on AmbigQA$^*$ are largely explained by its considerable proportion of near-zero entropy ground-truth samples.

Table 7: AUCROC scores for Gemma 3-12B for different uncertainty thresholds $\delta$ across all estimators

| Uncertainty Threshold $\delta$ | AU = 0 | | | | AU ≥ 0 | | | | | | | |
| | **TriviaQA** | | | | **MAQA$^*$** | | | | **AmbigQA$^*$** | | | |
| | SE | MSP | SAR | IP | SE | MSP | SAR | IP | SE | MSP | SAR | IP |
|---|---|---|---|---|---|---|---|---|---|---|---|---|
| $\delta = \log(1.5)$ | 0.95 | 0.89 | 0.92 | 0.94 | 0.54 | 0.52 | 0.58 | 0.61 | 0.78 | 0.73 | 0.77 | 0.78 |
| $\delta = \log(2)$ | 0.93 | 0.87 | 0.90 | 0.92 | 0.56 | 0.53 | 0.60 | 0.63 | 0.75 | 0.71 | 0.74 | 0.75 |
| $\delta = \log(3)$ | 0.90 | 0.85 | 0.89 | 0.89 | 0.58 | 0.56 | 0.62 | 0.65 | 0.73 | 0.70 | 0.73 | 0.73 |

### A.2 INTERNAL REPRESENTATIONS

In addition to our regression experiments, we train classifiers to predict a binary certainty label $y \in \{0, 1\}$. The label is obtained by thresholding the true epistemic uncertainty at $\delta = \log(2)$, consistent with the procedure used in the previous experiments. We train linear probes $\sigma(\langle\theta, h^l\rangle)$, where $\sigma$ denotes the sigmoid function, and 2 Layer MLPs to distinguish between low and high epistemic uncertainty samples. Figure 6 and table 8 shows the result for the MLP classification probes. Similarly, as in the regression case, we see a significant gap between performance on the different aleatoric uncertainty regimens.

Table 8: $AUCROC$ for probes with certainty threshold $\delta = \log(2)$.

| Model | AU = 0 | | | | | |
| | TriviaQA | | MAQA | | AmbigQA | |
| | Linear | MLP | Linear | MLP | Linear | MLP |
|---|---|---|---|---|---|---|
| Llama 3.1-8B | 0.76 | 0.79 | 0.64 | 0.65 | 0.64 | 0.66 |
| Gemma 3-12B | 0.80 | 0.83 | 0.69 | 0.69 | 0.64 | 0.69 |
| Qwen 2.5-14B | 0.74 | 0.77 | 0.67 | 0.65 | 0.68 | 0.71 |

# B  IMPLEMENTATION DETAILS

## B.1  APPROXIMATIONS

**Approximation of** $p$    To estimate the probability $p(y)$ of a semantic class $y \in \mathcal{C}$, we sample $K$ answers $a_1, \ldots, a_K$ from the model and then cluster them into semantic classes using an auxiliary entailment model. The probabilities of each semantic class are then obtained by aggregating and normalizing the answer probabilities within each class:

$$p(y) \approx \frac{\tilde{p}(y)}{\sum_{j=1}^{|\mathcal{C}|} \tilde{p}(y_j)}, \quad \text{where} \quad \tilde{p}(y) = \frac{1}{K} \sum_{i=1}^{K} \mathbb{I}(a_i \in y) p(a_i), \quad a_i \sim p(a).$$

As $K \to \infty$, the approximation converges to the model's true semantic answer distribution. We use a higher number of samples $K = 30$ to ensure a reasonable approximation. Semantic clustering follows the procedure of Kuhn et al. (2023), employing a bi-directional entailment check with the *deberta-v2-xlarge-mnli* model He et al. (2021). Samples are drawn via multinomial sampling with the default temperature, top-p, and top-k settings of each model. This choice is deliberate, as different model families and versions (e.g., base vs. instruct) provide different defaults, and we aim to evaluate them under their most realistic production settings.

**Calculation of Epistemic Uncertainty** $KL(p^* \parallel p)$    The distribution $p^*$ defines probabilities over the set of semantically distinct correct answers. Since the model distribution $p$ is sampled and may be arbitrary, their supports need not coincide. Moreover, matching classes may also differ in surface form. As such, they need to be *aligned* to be able to calculate the epistemic uncertainty. As an example, consider:

$$p^* = \{\text{Heat} : 0.3, \text{Fuel} : 0.34, \text{Oxygen} : 0.36\}$$
$$p = \{\text{It's Heat} : 0.4, \text{Carbon} : 0.2, \text{Oxygen} : 0.4\}.$$

We construct a joint support set $\{\text{Heat}, \text{Fuel}, \text{Oxygen}, \text{Carbon}\}$, imputing missing values with 0 in $p^*$ and with $\epsilon = 0.01$ in $p$ to avoid undefined terms in the KL-divergence due to $\log(0)$. Using $\epsilon$ for the model distribution is justified, since in principle the model assigns non-zero probability to any possible sequence, making the support of $p^*$ always a subset of the support of $p$. To determine the common support set, we apply the same semantic clustering procedure used for estimating $p$, based on bidirectional entailment with *deberta-v2-xlarge-mnli* He et al. (2021).

## B.2  PREDICTIVE VARIATION ESTIMATORS

**Semantic Entropy (SE)**    For semantic entropy, we follow Kuhn et al. (2023) The method first estimates the semantic distribution $p$ as outlined in Section B.1 using $K$ samples, and then computes the entropy:

$$H(p) = -\sum_{i=1}^{|C|} p_i \log p_i.$$

**Maximum Sentence Probability (MSP)**    A simple yet effective estimator is the maximum sentence probability (MSP), defined as:

$$\text{MSP} = 1 - \max_a \ p(a \mid x),$$

where $p(a \mid x)$ is the probability assigned to answer $a$. Importantly, we do not compute $\max_y p(y \mid x)$ from the semantic distribution $p$ estimated above; instead, we directly perform beam search with 5 beams to identify the highest-probability answer. This approach is similar to a recent proposal by Aichberger et al. (2024a)

**Shifting Attention to Relevance (SAR)**   Instead of having hard clusters, SAR computes continuous semantic similarity scores to determine the importance of samples. Additionally, SAR mitigates the influence of irrelevant tokens by calculating the importance of each token on the semantics of the answer (Duan et al., 2024). We use the implementation of (Vashurin et al., 2025) using *cross-encoder/stsb-roberta-large* as the semantic similarity model and $K = 30$ samples.

**Iterative Prompting (IP)**   The proposed estimator (Yadkori et al., 2024) should not be confused with the traditional MI estimator (Depeweg et al., 2018). The core idea behind the method is based on the idea that if a model is epistemically certain, it is less likely to change its answer by the inclusion of a wrong answer in the input context. For a detailed explanation of this method, we refer to Yadkori et al. (2024). In our implementation, we limit the number of samples to $K = 10$. Conditional probabilities are obtained via teacher forcing and extracted explicitly from the model output. We use hyperparameters $\gamma_1 = \gamma_2 = 10^{-9}$ and employ the prompt schema shown in Prompt 3 to obtain the conditional probabilities.

### B.3   Internal Representation Estimators

**Activations**   We use the residual stream activations, evaluated at the final token position of the input sequence, i.e., immediately before the model begins generating the answer. This position captures the complete contextual representation of the question and is therefore a natural choice for probing. In our setting, answers are typically short, making the first generated token particularly important and further motivating this choice. We also experimented with probing MLP and attention activations, but observed no substantial differences.

**Models**   For linear baselines, we use ridge regression and logistic regression with default `scikit-learn` settings. For non-linear probes, we employ two-layer MLPs (hidden dimensions 256 and 128) with ReLU activations and the Adam optimizer, implemented via `scikit-learn`.

**Evaluation**   All probes are evaluated with 3-fold cross-validation. In both regression and classification, we stratify the splits by binarized epistemic uncertainty (threshold $\log(\delta) = 2$). Reported results are mean scores across folds, with standard deviations shown in the figures.

### B.4   Ensemble Estimator

As our ensemble-based estimator, we adopt the classical mutual information (MI) formulation (Depeweg et al., 2018). Specifically, we treat LLaMA-3.1 8B, Gemma-3 12B, and Qwen-2.5 14B as approximate posterior samples from different architectures. The MI is then computed as the expected KL divergence between each member's predictive distribution $p_i$ and the ensemble mean $\bar{p}$:

$$\mathrm{MI}(Y; \theta) \;=\; \frac{1}{3} \sum_{i=1}^{3} \mathrm{KL}(p_i \,\|\, \bar{p}), \quad \text{where} \quad \bar{p} = \frac{1}{3} \sum_{i=1}^{3} p_i.$$

As in the calculation of epistemic uncertainty, we align the distributions at the semantic level, following the exact procedure described in Section B.1.

### B.5   Inference Prompts

For base models, we employ few-shot prompts to guide the model toward producing answers in the desired format (Prompt 1). In contrast, instruct models are queried with a single instruction that specifies the expected answer style (Prompt 2).

Prompt 1: Prompt for base models.

```
Q: What is one planet in our solar system that has rings?
A: Saturn

Q: Name one programming language you know.
A: Python

Q: Who is one of the singers in the band ABBA?
A: Agnetha Faeltskog

Q: What is one color in the German flag?
A: Black

Q: {question}?
A:
```

Prompt 2: Prompt for instruct models.

```
Answer the following question with one word or phrase:
{question}?
```

Prompt 3: Prompt for MI estimator

```
A possible answer to the question {question} is {answer}.
Q: {question}?
A:
```

## C  DATASET CREATION

Our dataset construction process consists of the following steps:

**Question Rephrasing:** Each original question is reformulated to explicitly request exactly one specific answer. E.g.: *"What are the essential components of the fire triangle?"* →*"What is one essential component of the fire triangle?"*. This prevents the model from producing multiple answers in a single generation. The rephrasing is done with `gpt-4.1-mini`.

**Keyword Extraction:** To enable the co-occurrence search, we extract a main keyword for the co-occurrence search. The keyword can either be a single word, like the subject, or a phrase. Critically, the co-occurrence of the keyword and the answer should reliably indicate the presence of the fact in the retrieved document. This is a valid assumption in most cases, as Elsahar et al. (2018) shows that when only the subject and object of a subject-object-relation triple co-occur in text, the resulting triple is often also present. However, for our main dataset *Wikipedia English*, we take additional measures to enhance quality as explained Section C.1. The keyword extraction is done using `gpt-4.1-mini` with Prompt 5 - except for the proxy using The Pile, which employs entity linking.

**Co-occurrence Search:** For each question, we perform a co-occurrence search for each answer on the proxy corpora. The final ground-truth distribution $p^*(\cdot|q)$ for a given question $q$ is then obtained by the relative frequency of the individual answer counts to all answer counts. To reduce potential biases, we discard samples in which at least one candidate answer has zero counts. Due to this fact, using the different proxies *English Wikipedia*, *RedPajama-V1*, and *The Pile* can result in different samples in the final datasets.

### C.1  WIKIPEDIA ENGLISH

**Dataset curation**  We use the structured Wikipedia Wikimedia Enterprise (2024) dataset, and specifically the English version, which consists of all English article pages in a structured way. For each article, we are leveraging all data in the *sections* tag. For the co-occurrence search, we use Pyserini and build the search index locally Lin et al. (2021). To define what constitutes a document—i.e., how articles are chunked for indexing—we leverage the dataset's hierarchical structure: articles are organized into sections and subsections down to the level of individual paragraphs or

lists. We assume that relevant facts are contained at this lowest level, which represents a coherent unit of text. The average length of the resulting chunks is around 65 words, with the distribution following a power-law: fewer than 1% of the chunks exceed 300 words, while only a small number of outliers contain more than 2000 characters ($\approx$400 words). For such extreme outliers, we apply additional splitting at sentence boundaries. Importantly, apart from these rare cases, we keep the chunks intact and do not split them further, ensuring high recall of facts. Importantly, we also apply stemming to reduce words to their base forms, avoiding reliance on overly specific surface forms. The final index contains 65,069,586 documents.

**Co-occurrence counting**  In the retrieval step, we return all documents containing both the keyword and the candidate answer for a given question. Because the relationship between a question and its answer can be complex, relying on a single keyword often yields high recall but only moderate precision. For instance, consider the question *"Who is the founder of Apple?"*—one valid answer is *Steve Jobs*. If we extract *Apple* as the main keyword, then any fact expressing *"Steve Jobs founded Apple"* will naturally contain both *Steve Jobs* and *Apple*, which ensures high recall. However, the mere co-occurrence of *Steve Jobs* and *Apple* does not always capture the intended fact—for example, *"Steve Jobs was the CEO of Apple"*. Such cases reduce precision. Hence, to ensure high precision, we apply an entailment procedure. Given a retrieved document through the co-occurrence search, we pass it to an LLM to verify that the fact is indeed present. For this step, we use *Gemma-3 12B Instruct* with the prompt shown in Prompt 4 and examples in Table 10. To keep the entailment step computationally feasible, we cap the number of retrieved documents per candidate answer at 1000—a threshold that we observe is rarely exceeded. The final number of samples for MAQA* is 468 and for AmbigQA* 2553 (Table 9).

Prompt 4: Prompt for entailment check.

```
You are an expert at verifying factual entailment. I.e., is the fact
present in the text?
Given the following TEXT and FACT, answer with "yes" if the FACT
follows from the TEXT, or "no" if it does not.

TEXT: {text}
FACT: {fact}
Answer:
```

## C.2 REDPAJAMA-V1

**Dataset curation**  The Infini-Gram API provides access to co-occurrence counts across a range of large-scale pre-training datasets Liu et al. (2024). We use *RedPajama-v1* Weber et al. (2024), which closely replicates the LLaMA pre-training corpus and includes a diverse set of data sources.

**Co-occurrence counting**  Similarly, as for Wikipedia English, we query for co-occurrences of the keyword with each candidate answer. For the Infini-Gram API we use the parameters *max_diff_tokens*= 100 and *max_clause_freq*= 50000. Since the underlying tokenizer (LLaMA 2) is sensitive to whitespaces for a keyword answer pair, we test all four combinations of including or removing a whitespace at the beginning of the keyword or answer. To obtain the final counts, we sum up the retrieved counts of the four different possibilities. Due to limited document access in Infini-Gram, we do not perform an entailment-checking phase. The final number of samples for MAQA* is 470 and for AmbigQA* 2331 (Table 9).

## C.3 THE PILE

**Dataset Curation**  In contrast to the previous two approaches, this method follows a different strategy for obtaining keywords and answers. It relies on entity linking, which identifies entities such as people, cities, or songs in both the question and the answer. The co-occurrence of a question entity with an answer entity is then retrieved from the Pile corpus Gao et al. (2020). Following the approach of Kandpal et al. (2023), we use the DBpedia Spotlight entity linker Daiber et al. (2013) to extract entities from questions and answers. To improve accuracy, each answer is appended to its corresponding question before entity linking. When multiple candidate entities are returned

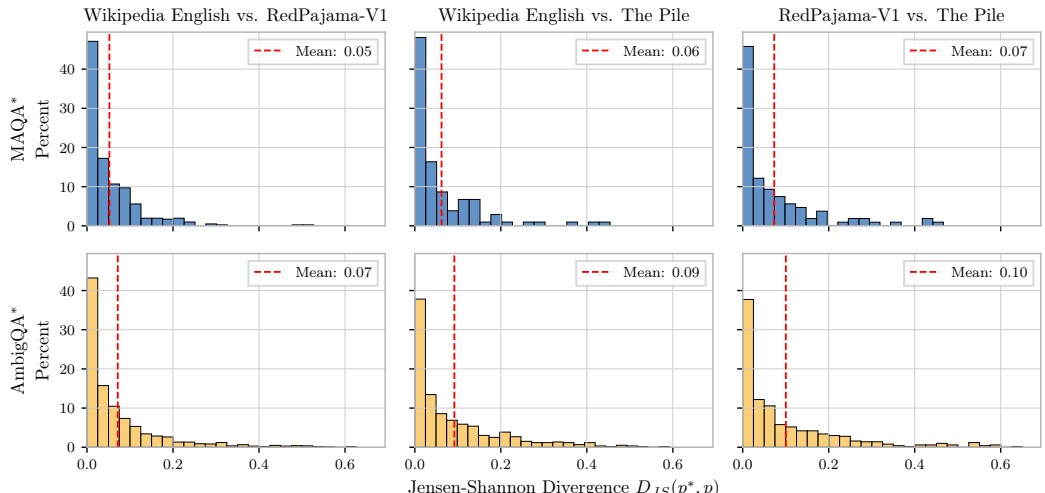

Figure 7: Comparison of retrieved ground-truth distribution $p^*$ using different strategies

for a question, we employ *Gemma-3 12B Instruct* to filter for the most relevant one. The linker's parameters are set to *confidence* = 0.4 and *support* = 1.

**Co-occurrence counting**  After obtaining the entity sets, we match them with pre-extracted entities from The Pile provided by Kandpal et al. (2023) to compute co-occurrence statistics. Similarly, as for RedPajama-V1, we do not perform an entailment-checking phase as we do not have access to the underlying documents.The final number of samples for MAQA$^*$ is 120 and for AmbigQA$^*$ 861 (Table 9).

## C.4  CHARACTERISTICS

Summary statistics are reported in Table 9. Compared to Wikipedia English, the other two strategies have access to a substantially larger pre-training corpus and therefore yield considerably higher counts. Nevertheless, the average entropies and their standard deviations remain in a similar range. As mentioned previously, we use *English Wikipedia* as our principal strategy since it is the most controlled method with entailment checking, ensuring high precision and high recall. As can be seen in Table 9, it also provides most samples on AmbigQA$^*$ and similarly many on MAQA$^*$ as RedPajama-V1. Using The Pile, in contrast, produces significantly fewer samples compared to the other two methods, as entity linking often can't find an entity in either question or answer, and hence such samples have to be discarded. To assess how well the estimated ground truths $p^*$ align across datasets, we compute the Jensen-Shannon divergence for all pairwise couplings on MAQA$^*$ and AmbigQA$^*$ (Figure 7). The Jensen-Shannon divergence is given by: $\text{JS}(p \,\|\, q) = \frac{1}{2}\left[\text{KL}(p \,\|\, m) + \text{KL}(q \,\|\, m)\right]$ ,where $m = \frac{1}{2}(p + q)$. It has the useful property of being symmetric, as we do not consider one strategy over the other as the truth. Overall, all strategies produce largely consistent ground truths, as reflected in the low average JS divergence and the characteristic power-law distribution (Figure 7).

Table 9: Summary statistics for $p^*$ estimation strategies: samples $n$, mean answer-counts, and mean entropies (mean ± std).

| $p^*$ **Estimation Method** | **MAQA$^*$** | | | **AmbigQA$^*$** | | |
|---|---|---|---|---|---|---|
| | **n** | $\overline{\textbf{Count}} \pm \textbf{SD}$ | $\overline{H} \pm \textbf{SD}$ | **n** | $\overline{\textbf{Count}} \pm \textbf{SD}$ | $\overline{H} \pm \textbf{SD}$ |
| Wikipedia English | 468 | 115.49 ± 178.63 | 1.11 ± 0.44 | 2553 | 55.81 ± 104.77 | 0.64 ± 0.33 |
| RedPajama-V1 | 470 | 143066.88 ± 1234461.86 | 0.99 ± 0.44 | 2331 | 1220812.64 ± 29688717.59 | 0.52 ± 0.35 |
| The Pile | 120 | 46281.74 ± 138909.80 | 0.97 ± 0.44 | 861 | 19881.87 ± 55441.02 | 0.60 ± 0.33 |

## D   PROOFS

**Proposition 1** (Non-Identifiability of Epistemic Uncertainty). *Let $K \geq 2$ and $\Delta^{K-1}$ be the probability simplex over $K$ classes. For any function $f : \Delta^{K-1} \to \mathbb{R}$ and any $p \in \Delta^{K-1}$, there exist $p_1^*, p_2^* \in \Delta^{K-1}$ such that*

$$\mathrm{KL}(p_1^* \| p) = 0 \quad and \quad \mathrm{KL}(p_2^* \| p) = -\log \min_i p_i \geq \log K,$$

*Thus, the model's prediction $p$, and consequently any function $f(p)$, can both indicate zero epistemic uncertainty or high epistemic uncertainty ($\geq \log(K)$).*

*Proof.* Fix $p \in \Delta^{K-1}$. Set $p_1^* := p$. Then $\mathrm{KL}(p_1^* \| p) = 0$. Let $j \in \arg\min_i \ p_i$ and define $p_2^* := \mathbf{1}[y = j]$. Then

$$\mathrm{KL}(p_2^* \| p) = -\log p_{\min}.$$

Thus, for the same $p$, EU can be 0 or large, while $f(p)$ is fixed. $\qquad\square$

**Proposition 2** (High MI $\not\Rightarrow$ High EU). *Let $K \geq 2$ and $\Delta^{K-1}$ be the probability simplex over $K$ classes. Let $\delta \in [0, \log K]$ be an arbitrary threshold on MI indicating uncertainty. Let $p_\theta$ be such that $\mathrm{MI}(\bar{p}; \theta) > \delta$ with $\bar{p} = \mathbb{E}_\theta[p_\theta]$. Then $p^* = \bar{p} \in \Delta^{K-1}$ results in true epistemic uncertainty $\mathrm{KL}(p^* \| \bar{p}) = 0$.*

*Proof.* Let there be a distribution $p_\theta$ such that $\mathrm{MI}(\bar{p}, \theta) > \delta$ for some $\delta \in [0, \log K]$. Since the probability simplex $\Delta^{K-1}$ is convex and $p_\theta \in \Delta^{K-1}$, the expected $\bar{p} = \mathbb{E}_\theta[p_\theta] \in \Delta^{K-1}$. Therefore, if the true distribution $p^* = \bar{p}$, the EU $\mathrm{KL}(p^* \| \bar{p})$ is trivially 0. Thus, for any arbitrary estimate of EU through MI, there exists a true distribution with zero EU. $\qquad\square$

**Proposition 3** (Zero aleatoric uncertainty implies EU is NLL).

$$H(p^*) = 0 \implies EU = -\log(p(y = y^*))$$

*Proof.* If $H(p^*) = 0$, then $p^*(y) = \mathbf{1}[y = y^*]$. From this it follows:

$$EU = KL(p^* \| p) = -\sum_{y \neq y^*} 0\log(p(y)) - \log(p(y = y^*)) = -\log(p(y = y^*))$$

$$\square$$

**Theorem 1** (High Entropy $\Rightarrow$ High EU). *Let there be $K \geq 2$ classes and $\delta \in [0, \log K]$ be a threshold on the entropy indicating uncertainty. Furthermore, let $\alpha_\delta$ be the maximal possible probability on some class s.t. $H(p) \geq \delta$. Then the epistemic uncertainty with $H(p) \geq \delta$ is at least:*

$$EU = \mathrm{KL}(p^* \| p) \geq -\log \alpha_\delta.$$

*Proof.* We first define $\alpha_\delta$ mathematically and how to obtain it.

$$\alpha_\delta = \max\left\{ \max_j p_j \ : \ H(p) \geq \delta \right\}, \qquad \delta \in [0, \log K].$$

Let $H_{max}(\alpha) = -\alpha \log \alpha - (1-\alpha) \log \frac{1-\alpha}{K-1}$. This is the maximum entropy achievable by a distribution whose largest class probability is $\alpha \in [1/K, 1]$. Then $\alpha_\delta$ is the solution of $H_{max}(\alpha) = \delta$. Now we seek the lowest possible $EU = -log(p(y = y^*))$ under the constraint $H(p) \geq \delta$. This exactly occurs if the maximal possible probability $\alpha_\delta$ is on the correct class and hence $EU = -\log(p(y = y^*)) \geq -\log(\alpha_\delta)$ $\qquad\square$

**Theorem 2** (Low Entropy $\Rightarrow$ Low EU with High Probability). *Let there be $K \geq 2$ classes and $\delta \in [0, \log 2]$ be a threshold on the entropy indicating uncertainty. Furthermore let $\bar{\mathcal{L}} = \mathbb{E}_{(x,y)}[-\log p_y]$ be the model's average loss and $\gamma_\delta$ be the minimal maximal confidence in a prediction $p$ s.t $H(p) \leq \delta$. Then the probability that the epistemic uncertainty with $H(p) \leq \delta$ will be less than $-\log(\gamma_\delta)$ satisfies:*

$$\mathbb{P}(EU \leq -\log(\gamma_\delta) \mid H(p) \leq \delta) \geq 1 - \frac{\bar{\mathcal{L}}}{-\log(1 - \gamma_\delta) * \mathbb{P}(H(p) \leq \delta)}$$

*Proof.* We first define $\gamma_\delta$ mathematically and how to obtain it:

$$\gamma_\delta \ = \ \min\Big\{ \max_j p_j \ : \ H(p) \leq \delta \Big\}, \qquad \delta \in [0, \log 2],$$

Denote $H_B(\gamma) = -\gamma \log \gamma - (1 - \gamma) \log(1 - \gamma)$ as the binary entropy function. Then $\gamma_\delta$ is the solution of $H_B(\gamma) = \delta$ for $\gamma \in [1/2, 1]$ and we can now proceed:

$$\mathcal{L} = \mathbb{E}_{(x,y^*)}\big[-\log p_{y^*}\big] \tag{3}$$

$$= \mathbb{E}_{(x,y^*)}\big[-\log p_{y^*} \mid H(p) \leq \delta\big]\mathbb{P}(H(p) \leq \delta) \tag{4}$$

$$+ \mathbb{E}_{(x,y^*)}\big[-\log p_{y^*} \mid H(p) > \delta\big]\mathbb{P}(H(p) > \delta)$$

$$= \mathbb{E}_{(x,y^*)}\big[-\log p_{y^*} \mid H(p) \leq \delta \cap \arg\max p \neq y^*\big]\mathbb{P}(H(p) \leq \delta \cap \arg\max p \neq y^*) \tag{5}$$

$$+ \mathbb{E}_{(x,y^*)}\big[-\log p_{y^*} \mid H(p) \leq \delta \cap \arg\max p = y^*\big]\mathbb{P}(H(p) \leq \delta \cap \arg\max p = y^*)$$

$$+ \mathbb{E}_{(x,y^*)}\big[-\log p_{y^*} \mid H(p) > \delta\big]\mathbb{P}(H(p) > \delta)$$

$$\geq -\log(1 - \gamma_\delta)\mathbb{P}(H(p) \leq \delta \cap \arg\max p \neq y^*) - \log(\alpha_\delta)\mathbb{P}(H(p) > \delta) \tag{6}$$

Where we use in 4 the law of total expectation to separate into high and low entropy predictions. In 5, we further partition the space of low entropy predictions into correct and incorrect ones. Lastly, in 6, we bound the expectation values. High entropy predictions occur at least loss $-\log(\alpha_\delta)$ according to theorem 1. Low entropy predictions that are incorrect will have maximally $1 - \gamma_\delta$ mass on an *correct* class and as such occur at least $-\log(1 - \gamma_\delta)$ loss. Rearranging terms and substituting $\mathbb{P}(H(p) > \delta) = 1 - \mathbb{P}(H(p) \leq \delta)$ yields

$$\mathbb{P}(H(p) \leq \delta \cap \arg\max p \neq y^*) \leq \frac{\mathcal{L} + (1 - \mathbb{P}(H(p) \leq \delta))\log(\alpha_\delta)}{-\log(1 - \gamma_\delta)}$$

Dividing by $\mathbb{P}(H(p) \leq \delta)$ we finally get the conditional bound:

$$\mathbb{P}(\arg\max p \neq y^* \mid H(p) \leq \delta) \leq \frac{\mathcal{L} + (1 - \mathbb{P}(H(p) \leq \delta))\log(\alpha_\delta)}{-\log(1 - \gamma_\delta)\mathbb{P}(H(p) \leq \delta)}$$

which can be rewritten to obtain an upper bound as:

$$\mathbb{P}(\arg\max p = y^* \mid H(p) \leq \delta) \geq 1 - \frac{\mathcal{L} + (1 - \mathbb{P}(H(p) \leq \delta))\log(\alpha_\delta)}{-\log(1 - \gamma_\delta)\mathbb{P}(H(p) \leq \delta)} \tag{7}$$

$$= 1 - \frac{\mathcal{L}}{-\log(1 - \gamma_\delta)\mathbb{P}(H(p) \leq \delta)} \tag{8}$$

$$+ \frac{-\log(\alpha_\delta)(1 - \mathbb{P}(H(p) \leq \delta))}{-\log(1 - \gamma_\delta)\mathbb{P}(H(p) \leq \delta)} \tag{9}$$

Realizing that $-\log(p_{y^*}) \leq -\log(\gamma_\delta) \iff \arg\max p = y^*$ - since $\gamma_\delta$ is the minimum possible maximum probability - we get:

$$\mathbb{P}(\log p_{y^*} \leq -\log(\gamma_\delta) \mid H(p) \leq \delta) \geq 1 - \frac{\mathcal{L}}{-\log(1 - \gamma_\delta)\mathbb{P}(H(p) \leq \delta)} \tag{10}$$

$$+ \frac{-\log(\alpha_\delta)(1 - \mathbb{P}(H(p) \leq \delta))}{-\log(1 - \gamma_\delta)\mathbb{P}(H(p) \leq \delta)} \tag{11}$$

Abbreviating $-\log p_{y^*}$ as *epistemic uncertainty* EU and simplifying by leaving out the second term, we obtain the bound stated in the theorem.

$$\mathbb{P}(EU \leq -\log(\gamma_\delta) \mid H(p) \leq \delta) \geq 1 - \frac{\mathcal{L}}{-\log(1 - \gamma_\delta) * \mathbb{P}(H(p) \leq \delta)} \tag{12}$$

$\square$

## D.1 Non-trivial Aleatoric Uncertainty

When constraining $H(p^*) = 0$, we implicitly restrict $p^*$ to be a an indicator vector over one of the $K$ classes. As shown in Theorems 1 and 2, this setting allows for informative bounds on epistemic uncertainty. However, this is only one case. Consider instead the situation where $p^*$ is known exactly. While this assumption is unrealistic (since complete knowledge of $p^*$ makes estimation redundant), it helps to illustrate non-trivial aleatoric uncertainty. For example, if $p^*$ is uniform, we obtain maximal aleatoric uncertainty with $H(p^*) = \log K$. However, we can, in fact, exactly determine the epistemic uncertainty:

$$EU = KL(p^* \,||\, p) = \sum_y \frac{1}{K} \log(\frac{1}{Kp(y)}) = -\log(K) - \frac{1}{K} \sum_y \log(p(y))$$

Similarly when relaxing the constraint slightly to allow $p^*$ be a high entropy distribution (e.g., $H(p^*) \in [\log K - \epsilon, \log K]$) estimation of epistemic uncertainty using $H(p)$ should work reasonably: low predictive entropy necessarily implies high epistemic uncertainty, whereas high predictive entropy indicates lower epistemic uncertainty.

These illustrations clarify what we mean by *non-trivial* aleatoric uncertainty: cases where no strong restrictions on $H(p^*)$ are imposed. This is the typical regime in realistic applications, since constraining $H(p^*)$ would require prior knowledge about the ambiguity structure of the task itself. This is especially the case in many linguistic problems, as a specific language task can have an arbitrary, ambiguous structure.

Table 10: Examples of entailment check in the co-occurrence pipeline for Wikipedia English

| Idx | Question | Keyword | Answer | Positive Example | Negative Example |
|---|---|---|---|---|---|
| 72 | Who was the recipient of the Bharat Ratna award when it was first awarded? | Bharat Ratna | ['C. Rajagopalachari'] | article rajaji national park, section abstract: rajaji national park was named after c. rajagopalachari (rajaji), a prominent leader of the freedom struggle, the first and last governor-general of independent india and one of the first recipients of india's highest civilian award, bharat ratna (in 1954). | article central college, bangalore, section notable students: bharat ratna sir m. visvesvaraya, harshavardhan mudaliar, prof in english, bharat ratna c. rajagopalachari, bharat ratna c. n. r. rao, indian chemist, shivakumara swami, pusapati vijayarama gajapati raju, maharaja of vizianagaram, h. narasimhaiah, guruswami mudaliar, hospet sumitra, n. santosh hegde, justice, navaratna rama rao, leading administrator, author and founder of the sericulture department, n. s. subba rao, maya rao (1928-20... |
| 186 | What is one specific type of agricultural product the Wachau Valley is known for? | Wachau Valley | ['grapes'] | article wachau, section wine: the wachau valley is well known for its production of apricots and grapes, both of which are used to produce specialty liquors and wines. the wine district's rolling vineyards produce complex white wines. wachau is a source of austria's most prized dry rieslings and grüner veltliners, some of the best from the steep stony slopes next to the danube on which the vines are planted. the temperature variation in the valley between day and cold nights has a significant ro... | No negative example found |
| 198 | What is the name of one Unforgivable Curse from the Harry Potter books? | Unforgivable Curses | ['Imperio'] | article imperio, section abstract: imperio, a curse in the harry potter series (see magic in harry potter#unforgivable curses), imperio (band), austrian band | No negative example found |
| 205 | Who was one of the main cast members of 'The Big Valley' TV show? | The Big Valley | ['Barbara Stanwyck'] | article the big valley, section reception : popularity: in the comedy film airplane! (1980), the wacky air traffic controller johnny, played by stephen stucker, paid homage to big valley ' s penchant for big drama in one of his many asides. after lloyd bridges ' character frets about a pilot who cracked under pressure, johnny says: "it happened to barbara stanwyck!" and "nick, heath, jarrod – there's a fire in the barn!" the big valley also has seeped into the darker cinematic subconscious. in b... | article peter breck, section career : after the big valley: on january 20, 1990, while teaching at the drama school, breck was notified of barbara stanwyck's death. she requested no funeral nor memorial. |
| 297 | Which stadium did the New Orleans Saints use for their home games in the seasons following Hurricane Katrina in 2005? | New Orleans Saints | ['Alamodome'] | article tom benson, section biography : new orleans saints : saints relocation controversy: when it became clear that hurricane katrina 's extensive damage to new orleans and the superdome would make it impossible for the saints to play there in 2005, the team temporarily relocated its operations to san antonio and began negotiations to play home games at the alamodome. (the saints, after discussions with the nfl and louisiana state university, eventually agreed to play one "home" game at giants... | article 2001 minnesota vikings season, section preseason : game summaries : week 1: at new orleans saints: at alamodome, san antonio, texas |

Prompt 5: Prompt for keyword extraction.

```
You are a keyword extraction assistant helping to identify the
keywords in a question for a co-occurrence search.
The goal is to check how often the answer to a specific question
(fact) appears in a text corpus.
To do this, you must identify the keywords in the question that are
needed to find the fact in the text corpus.

Your job is to analyze a question/answer pair and pull out:
- The minimal term(s) that, when paired with the known answer entities,
    reliably locate the same fact in a text corpus.
- The goal is to have as few terms as possible while still being able to
    find the fact.

Guidelines:
- Extract the main keyword from the question that shrinks the search
    space.
  E.g., for a song title question, the main keyword is the title of the
    song.
- Extract additional keywords needed to find the fact in a text corpus.
  E.g., for a song title, additional keywords are the artist and album.
- The main keyword should be a single term or short phrase that captures
    the essence of the question.
- Additional keywords should be a short list of terms (not too long).

Return exactly this JSON (no extra fields or explanation):

{
  "main_keyword": [string],
  "additional_keywords": [ string, .. ]
}

Example 1
Input:
Question: "Who were the writers of the song 'Tell Your Heart to Beat
    Again'?"
Answer:   "Bernie Herms, Mathew West, Randy Phillips"
Output:
{
    "main_keyword": ["Tell Your Heart to Beat Again"],
    "additional_keywords": ["writers"]
}

Example 2
Input:
Question: "What are the names of recognized dwarf planets in the solar
    system as of 2024?"
Answer:   "Ceres, Eris, Pluto, Makemake, Haumea"
Output:
{
    "main_keyword": ["dwarf planet"],
    "additional_keywords": ["solar system"]
}

Example 3
Input:
Question: "What is the legal age of marriage in the United States?"
Answer:   "18, 19, 21"
Output:
{
    "main_keyword": ["marriage"],
    "additional_keywords": ["legal age", "United States"]
}
```

```
Now process the following and produce **only** the JSON:

Question: "{question}"
Answer:   "{answer}"
```

## E  USAGE OF LARGE LANGUAGE MODELS

In this work, we used LLMs to polish and rephrase minor sentences of the paper.

