# OpenReview forum: "The Illusion of Certainty: Uncertainty quantification for LLMs fails under ambiguity"
_ICLR.cc/2026/Conference — ICLR 2026 Conference Withdrawn Submission_

### Official Review · Reviewer_MuH1 · 2025-10-31

**Soundness:** 2
**Presentation:** 2
**Contribution:** 2
**Rating:** 2
**Confidence:** 4

**Summary:**

This paper focuses on ambiguity and uncertainty quantification (UQ) in question answering. The authors first expand existing ambiguous QA datasets by constructing a “ground-truth” answer distribution, where the probability of each candidate answer reflects its frequency in the pretraining corpus. They then evaluate three categories of uncertainty quantification methods on these ambiguous QA datasets and demonstrate that these methods fail to produce reliable uncertainty estimates under ambiguity.

**Strengths:**

The idea of assigning a ground-truth distribution to ambiguous questions is novel. Prior work primarily focused on collecting ambiguous questions and their corresponding disambiguated versions, whereas this paper explores what the ground-truth distribution should be for ambiguous cases. Also, the problem setting (examining how UQ methods behave under question ambiguity) is relevant to the reliability of LLMs.

**Weaknesses:**

- **Questionable Motivation for the Proposed “Ground-Truth Distribution.”** The authors do not justify why the so-caled ground-truth distribution for ambigous questions is needed. From my perspective, the ambiguity is objective -- it is there, and there are multiple ways to understand an ambigous question, leading to multiple correct answers. What we need is the clarifications for the ambiguity, the disambiguations for the question, and the corresponding correct answers. It does not makes sense to me to create a probability distribution for the ground-truth label for an ambiguous question, where the probability mass for each candidate answer reflects its frequency in the pre-training corpus. I do not think it make sense to define the ground-truth label in this way. LLMs should first recognize there is ambiguity and then provide different answers correspondingly, without assigning a ratio/weight for each answer.
- **Redundant and Trivial Sections.** Several sections present results that are well-known and widely accepted in the community, without offering new insight. Specifically,
  - Section 3:The claim that “UQ methods work under zero aleatoric uncertainty” is straightforward and has already been thoroughly addressed in prior works  [1-3]. Reproducing this observation empirically and theoretically seems unnecessary and adds little contribution.
  - Section 5: The analysis of “why seemingly robust estimators fail once AU is non-trivial” is also trivial. The paper itself notes that “a high-entropy prediction no longer necessarily indicates high epistemic uncertainty, as the entropy may also arise from inherent ambiguity in the ground truth.” I think formalizing it both empirically and theoretically does not advance understanding.
- **Lack of Substantive Contribution or Solution.** The paper identifies an obvious problem that existing UQ methods fail under ambiguity. However, it offers no new method or framework to address it. The conclusion (“current UQ methods are ineffective under ambiguity”) is expected and unsurprising. As a result, the paper reads more as a descriptive survey or negative result without sufficient theoretical or methodological contribution.

[1] Kuhn, Lorenz, Yarin Gal, and Sebastian Farquhar. "Semantic uncertainty: Linguistic invariances for uncertainty estimation in natural language generation." arXiv preprint arXiv:2302.09664 (2023).

[2] Cole, Jeremy R., et al. "Selectively answering ambiguous questions." arXiv preprint arXiv:2305.14613 (2023).

[3] Shavindra Jayasekera, I., et al. "Variational Uncertainty Decomposition for In-Context Learning." NeurIPS, 2025.

**Questions:**

Please refer to the weakness

---

### Official Review · Reviewer_sxrJ · 2025-10-31

**Soundness:** 3
**Presentation:** 3
**Contribution:** 2
**Rating:** 4
**Confidence:** 4

**Summary:**

The paper revisits uncertainty quantification for LLMs under conditions of ambiguity. Indeed, current UQ methods are typically benchmarked on datasets that do not contain any ambiguity. For example, on QA tasks, there is exactly one correct answer to each question. The current work drops this assumption and proposes two new benchmarks called MAQA* and AmbiQA*, both being QA datasets, such that the questions have multiple correct answers. Consequently, the empirical evaluation demonstrates that the classical UQ estimators nearly fail on these ambiguous datasets. This is an important result because many realistic use-cases are inherently ambiguous.

**Strengths:**

- Uncertainty quantification for LLMs is a hot topic and therefore advances in this field are definitely warranted.

- The paper sheds new light on UQ especially in the case of ambiguous QA tasks. It also motivates new research on developing UQ estimators that are sensitive to ambiguity.

**Weaknesses:**

- The results aren't that surprising. Recent work that looked at uncertainty quantification for the text-to-SQL task which is inherently ambiguous (namely, for a given natural language query there could be multiple correct SQL queries) already observed significant degradation in AUROC performance for several UQ estimators (see [1,2]).

[1] Bhattacharjya et al. SIMBA UQ: Similarity-Based Aggregation for Uncertainty Quantification in Large Language Models. EMNLP 2025.

[2] Bhattachariya et al. Consistency-based Black-box Uncertainty Quantification for Text-to-SQL. NeurIPS Workshop 2024.

**Questions:**

I suppose text-to-SQL would be a practical yet inherently ambiguous generative task. How would your approach extend to this task?

---

### Official Review · Reviewer_WuFW · 2025-11-01

**Soundness:** 3
**Presentation:** 2
**Contribution:** 3
**Rating:** 8
**Confidence:** 3

**Summary:**

This paper considers uncertainty estimation in the case of ambiguous questions, as defined in a specific manner by the work. The paper is conceptual in nature and slightly different from other work in the space, presenting some results and related insights as to why some existing UQ methods in the literature perform well for standard QA benchmarks with no ambiguity (like TriviaQA). The results rely on decomposing total uncertainty into epistemic and aleatoric components, as defined as cross entropy between the true distribution and predicted distribution over semantic classes of LLM output. The paper proposes datasets with ambiguous questions by modifying a couple of existing ones in the literature, where the main contribution is to provide them with ground truth distributions for the sort of analysis they need to perform. Evidence is provided in a manner that is both theoretical and empirical.

**Strengths:**

I think the paper makes some important contributions to understanding when some of the current UQ methods for LLMs perform well and why -- although I think the framing is still quite specific in that only a particular type of task and general setup (like using semantic classes to categorize output) is considered.

In my view, the insights are useful for the field of UQ for LLMs, novel enough to warrant interest from the broader community. I’m not sure about the practical value of the new datasets but I appreciate the effort and consider it a contribution as well. Overall, I think this is a strong paper.

**Weaknesses:**

I find the work limiting in some ways. The authors use a definition of total uncertainty that is not widely accepted and relies on a ground truth distribution p^*. In my view, they do not provide enough justification for this definition, choosing to merely cite a couple other papers. So while the work is useful and conceptually interesting, not enough is mentioned about the assumptions and cumbersome nature of setup. This is related to creation of the benchmark, which the authors discuss briefly in the limitations section, to their credit.

Another weakness is the lack of sufficient literature that is cited, both around uncertainty estimation and ambiguity. I suggest expanding the related work section. At the very least, I recommend including some review papers to show that the field is much broader than what is specifically cited and used as baselines.

**Questions:**

Some other comments/questions follow:

What are the probabilities mentioned on line 39? What are these probabilities over? Isn’t this a system decision? The authors likely mean some distribution over data. This is a key aspect that is not covered sufficiently in the paper and only mentioned later when the proposed datasets are described.

Some of the statements in the paper feel inappropriately strong, like in lines 83 – 86; several real problems do not have the sort of ambiguity in scope of this work.

How does one practically ascertain the number of possible classes over which an LLM produces as output? Does this not depend on the number of generated samples? How does this influence the results, if at all?

Please explain the choice for defining total uncertainty as done in equation (1). I recommend adding more detail in the paper.

In Section 2.1, please explain how to compute true EU. Mention that p^* is needed.

Where is Proposition 3, mentioned in line 156?

Minimal maximal confidence in line 192 is confusing. Explain what the min and max are over.

Explicitly mention the term in the inequality of Theorem 2 that represents the model making confident predictions. It would be helpful to explain how the probability term on the right hand side is about confident predictions.

There is a major typo in line 211 – it should be low predictive entropy.

Please fix issues with the citation style in many places in the paper. I also noticed several minor typos, like in lines 441 and 447.

---

### Author Response · Authors · 2025-11-20

Thank you for taking the time to review our paper. We really appreciate it. However, after internal discussions, we believe the paper can benefit from improvements beyond the rebuttal period. We have therefore decided to withdraw the submission.

---

### Note · Authors · 2025-11-20

**Comment:**

Thank you for taking the time to review our paper. We really appreciate it. However, after discussions, we believe the paper can benefit from revisions beyond the rebuttal period. We have therefore decided to withdraw the submission.

**Withdrawal Confirmation:**

I have read and agree with the venue's withdrawal policy on behalf of myself and my co-authors.